# Clinical Implications of Difference in Antigenicity of Different Botulinum Neurotoxin Type A Preparations: Clinical Take-Home Messages from Our Research Pool and Literature

**DOI:** 10.3390/toxins12080499

**Published:** 2020-08-04

**Authors:** Sara Samadzadeh, Beyza Ürer, Raphaela Brauns, Dietmar Rosenthal, John-Ih Lee, Philipp Albrecht, Harald Hefter

**Affiliations:** Department of Neurology, University Hospital of Düsseldorf, Moorenstrasse 5, D-40225 Düsseldorf, Germany; sara.samadzadeh@yahoo.com (S.S.); beyza.uerer@uni-duesseldorf.de (B.Ü.); montanabrauns@t-online.de (R.B.); rosenthal@med.uni-duesseldorf.de (D.R.); john-ih.lee@med.uni-duesseldorf.de (J.-I.L.); philipp.albrecht@med.uni-duesseldorf.de (P.A.)

**Keywords:** low antigenicity, incobotulinum toxin, neutralizing antibodies, secondary treatment failure, complex proteins: botulinum toxin type A, cervical dystonia

## Abstract

The three different botulinum toxin type A (BoNT/A) preparations being licensed in Europe and the U.S. differ in protein content, which seems to be a major factor influencing the antigenicity of BoNT/A. In the present study, several arguments out of our research pool were collected to demonstrate that the clinical response and antigenicity were different for the three BoNT/A preparations: some results of (1) a cross-sectional study on clinical outcome and antibody formation of 212 patients with cervical dystonia (CD) being treated between 2 and 22 years; (2) another cross-sectional study on the clinical aspects and neutralizing antibody (NAB) induction of 63 patients having developed partial secondary treatment under abobotulinum (aboBoNT/A) onabotulinumtoxin (onaBoNT/A) who were switched to incobotulinumtoxin (incoBoNT/A) in comparison to 32 patients being exclusively treated with incoBoNT/A. These results imply that (1) the presence of NAB cannot be concluded from the course of treatment, that (2) an increase in the dose and variability of outcome with treatment duration indicates the ongoing induction of NABs over time, that (3) the higher protein load of BoNT/A goes along with a higher incidence and prevalence of NAB induction and that (4) the best response to a BoNT/A is also dependent on the protein load of the preparation.

## 1. Introduction

For a variety of indications, the injections with botulinum neurotoxin type A (BoNT/A) have become the treatment of choice. Clinical efficacy follows a typical course. A few days after injection, the clinical effect of a BoNT/A injection increases rapidly, reaches a maximum after about 3 to 5 weeks, and decreases again. The duration of action depends on disease entities, patients, doses, and the precision of injections. When a further injection is performed before the effect of the previous injection has fully declined, an even better result can be observed than after the previous injection has fully vanished. Therefore, with repetitive injections, a progressive improvement can be achieved, which may have an even better effect than the 3–5 weeks. However, with repeated injections, the risk of antibody (AB) induction will increase [1]. 

In most cases, the mode of application is traumatic, which implies that the dendritic cells are activated and the B-cells detect fragments of the BoNT/A complex. Therefore, on a long-term basis, the induction of ABs can hardly be avoided. Depending on the epitopes, ABs attack and therefore, they may reduce the biological function of BoNT/A and cause partial secondary treatment failure (PSTF) with a shortening of the duration of action. The induction of neutralizing antibodies (NABs) and the development of PSTF may occur early in the course of treatment. The development of complete secondary treatment failure (CSTF), not only with the shortening of the duration of action, but also the complete abolishment of the 3–5 week effect has been observed after three injection cycles. On the other hand, it may last up to 300 months or more than 100 injection cycles until CSTF develops [2,3,4,5].

Risk factors for the development of a PSTF are of a high dose per treatment session, a long duration of treatment, booster injections, and the BoNT/A preparation used for treatment. In several studies on long-term treatment with BoNT/A, an increase in dose per session with the duration of treatment is observed. In patients with NABs, the paralysis time of the mouse hemidiaphragm test (MHDA) is significantly correlated with dose. Therefore, the MHDA is an appropriate instrument for the potency testing of different botulinum toxin drugs. The influence of booster injections has been mentioned as risk factors from the very beginning, however, detailed information is missing, probably because in most patients, booster injections are avoided [6,7,8,9,10,11,12,13].

The influence of the BoNT/A preparation on PSTF and NAB induction is very well known since the introduction of the “new” Botox^®^ [14,15,16,17]. The comparison of the three BoNT/A preparations being licensed in Europe and the U.S. reveals that there are also relevant differences between the complex protein-free incobotulinumtoxin (incoBoNT/A) (Xeomin^®^, Merz Pharmaceuticals, Germany) and the complex protein containing abobotulinum (aboBoNT/A) (Dysport^®^, Ipsen Pharmaceuticals, France) and onabotulinumtoxin (onaBoNT/A) (Botox^®^, Allergan, USA) [18,19]. The protein content of the aboBoNT/A and onaBoNT/A preparation were 4.87 and 5 ng per vial of 100 units, respectively. In the incoBoNT/A preparation, not only the complex proteins were removed, but also the biologically inactive fragments of the BoNT/A molecule. This leads to the very low protein content of 0.44 ng of the incoBoNT/A preparation [19].

There is some evidence for a difference in antigenicity of these three BoNT/A-preparations. In several studies, it has been reported that in patients being long-term treated for different indications exclusively with incoBoNT/A, no induction of NABs could be detected by the means of the sensitive MHDA [2,5,20]. Furthermore, the decrease in NAB-titres being induced after abo- or onaBoNT/A therapy after the switch to incoBoNT/A underlines the low antigenicity of incoBoNT/A [21], and a higher prevalence of MHDA-positive patients was observed after the abo- or onaBoNT/A monotherapy than after incoBoNT/A monotherapy [2,8].

In the present paper, further evidence is presented that the difference in antigenicity of the three BoNT/A preparations has relevant clinical implications. We will focus on patients with cervical dystonia (CD), since clinical scores are available for CD to semi-quantify disease severity [8], but all the results presented here can in principle be transferred to other disease entities [22].

## 2. Results

### 2.1. Clinical Hints for the Presence of NABs in an Individual Patient

When BoNT/A therapy is started in a patient in our department, we instruct the patient to assess the treatment effect by the daily scoring of the remaining severity of CD, in per cent of the untreated situation, just before the first BoNT/A injection on a visual analogue scale (VAS 0–100; 0  =  no more symptoms, 100  =  CD severity just as bad as before start of BoNT-therapy) (Figure 1). 

In addition, the treating physician scores the disease severity of CD by the means of the TSUI score [23] before each BoNT/A injection. A comparison of the patient’s assessment and physician´s scoring is helpful to document the efficacy of BoNT/A therapy.

In Figure 1A, a typical development of CD severity is documented in a patient with an initial TSUI score of 10 being treated with 300 MU incoBoNT/A. The patient´s assessment (left side) and physician´s scoring (right side) demonstrate that the severity of CD was gradually decreasing reaching a low plateau around 25% and a TSUI score of 2 after six injections. The patient and treating physician were satisfied. There was no clinical hint for the presence of NABs in this patient. The MHDA test was negative.

In Figure 1B, the temporal development of CD severity is presented for a patient with an initial TSUI score of 8 being treated with 250 MU onaBoNT/A. The patient´s assessment (left side) and the physician´s scoring (right side) show an initial gradual improvement reaching a plateau around 50% and a TSUI score of 4. The patient and treating physician were satisfied with the treatment effect. However, during a cross-sectional study, the NABs were detected in this patient with a moderate MHDA-titer of 5 mU/mL.

In Figure 1C, the temporal development of CD severity is presented for a patient with an initial TSUI score of 10 who was treated with 500 MU aboBoNT/A. After six injections, the patient´s assessment shows a very mild secondary worsening, and the scoring of the treating physician did not reveal significant changes. During a cross-sectional study, NABs were detected with a low MHDA titer of 1 mU/mL. 

In Figure 1D, the patient (left side) and treating physician (right side) had suspected a secondary treatment failure after an initial good response to 500 MU aboBoNT/A over five injection cycles. Although the dose was increased up to 1000 MU aboBoNT/A, the severity of CD continued to worsen. The MHDA confirmed an antibody-induced secondary treatment failure with a high titer (>10 mU/mL). 

These examples demonstrated that the patient´s assessment and the physician´s scoring of the outcome of BoNT/A therapy may be helpful in suspecting PSTF and NAB induction in a patient. However, the precise information on the prevalence of NABs can only be achieved by the rigorous testing of the entire study population (cross-sectional testing, [8,22]). In 2015, all the CD patients in our BoNT/A ambulance who were treated for at least 2 years, and did not have interrupted BoNT/A therapy for more than two cycles, were asked for participation in a cross-sectional MHDA-test study. Out of the 222 recruited patients, finally, 212 patients were tested. When the patients were split-up into subgroups depending on the severity of CD by means of the TSUI score, exceptional patients with a positive MHDA test, but a TSUI score of zero, were detected [8]. The percentage of patients with NABs in the lowest TSUI score group 0–2 was 12.5%. The percentage of a positive MHDA-test significantly increased (non-linear) with the TSUI score, with a dose (uDU) and with a duration of treatment (Figure 2 A–C).

### 2.2. Clinical Hints for the Presence of NAB Positive Patients in a Cohort of Patients

When the BoNT/A treatment was started in a cohort of CD patients with a mean TSUI score (MV0) and a standard deviation (SD0), the number of primary non-responders is usually small [5,24]. If all responding patients would have experienced an improvement of more than 10% before the next injection was applied, the TSUI mean value (MV1) would have decreased by at least 10%. It is also highly likely that the standard deviation (SD1) would have decreased. Thus, it has to be expected that both the TSUI mean values and the corresponding SDs gradually decrease with the number of applied injections. However, this does not seem to be the case in clinical practice.

The study of Kessler et al. [25] presented mean values and standard deviations of the TSUI scores of the first 12 aboBoNT/A injections in a large cohort of CD patients (n = 303). Although the mean TSUI scores gradually decreased, the corresponding SDs remained constant. These data are presented in Figure 3A. In the aforementioned cross-sectional study [8] analyzing 212 CD patients who were treated for between 2 and 22 years, the TSUI score did not correlate with the duration and the mean TSUI scores of the treatment segments of 2 years were constant, but the corresponding standard deviations gradually increased (Figure 3B). The calculation of the coefficient of variability (SD/MV*100) shows a logarithmic increase in both studies which nicely fit together (Figure 3C) demonstrating that the decrease in the MVs and constancy of the SDs during the first 12 injections in the Kessler study [25] follow the same rule as the constancy of the MVs and the increase in SDs in the Hefter study [8]. This is probably the result of the worsening of CD severity in the few patients developing NAB-induced PSTF during the course of treatment, which is the driving force to keep the variability of response high in a cohort of CD patients otherwise responding well.

### 2.3. Incidence and Prevalence of NAB-Induction Is Different for the Three BoNT/A Preparations

In the aforementioned cross-sectional study [8], those patients were selected in different subgroups who had been treated exclusively with aboBoNT/A (ABO group; n = 128) or onaBoNT/A (ONA group; n = 36). We compared these two groups to a recently published group of CD patients (n = 32) who were exclusively treated with incoBoNT/A (INCO group) [2]. In the INCO group, no patient had developed NABs (INCO group: the prevalence of MHDA-positive patients = 0%), in the ONA group, two patients were MHDA-positive (ONA group: the prevalence of MHDA-positive patients = 5.6%) and in the ABO group, 17 patients (ABO group: the prevalence of MHDA-positive patients = 13.3%). Since the mean duration of treatment was different for these three groups, differences in the incidences of the three subgroups resulted (Figure 4A).

### 2.4. Best Outcome Is Different for the Three BoNT/A Preparations

In a recent study, the best outcome (BTSUI) after the initiation of BoNT/A therapy in CD patients who developed PSTF years later was compared to the best outcome of CD patients who were exclusively treated with incoBoNT/A. The patients were split-up according to the preparation used at the onset of BoNT/A therapy yielding in the abo and onaBoNT/A group. The initial TSUI score (ITSUI) at the onset of BoNT/A was 8.74 +/- 3.67 in the abo- and 8.83 +/- 2.61 in the ona group. The mean of BTSUI was 3.75 after 1354 days of treatment, 3.83 after 1085 days of treatment in the abo- and ona group, respectively (Figure 4B). In the inco group, BTSUI was 1.71 and significantly better and was reached after a shorter duration of treatment (920 days) (Figure 4B). In the abo- and ona group SDs remained constant, whereas in the inco group, the SD significantly decreased (Figure 4C).

For each patient, the BTSUI was plotted against time to the best TSUI and connected with the corresponding ITSUI at day 0 (Figure 5). It is quite obvious that the BTSUI of the patients in the incoBoNT/A group (Figure 5B) was better and occurred earlier than in most of the patients in the abo- or onaBoNT/A group (Figure 5A).

## 3. Discussion

### 3.1. Clinical Hints for the Presence of NABs in an Individual Patient

The traumatic usual mode of BoNT/A application—repetitive injections—starts the pathway through dendritic cells, B-cells and T-cells to induce antibody formation against BoNT/A [26]. The detection of the BoNT/A by those cells, does not automatically imply that high titers of AB are induced with a relevant clinical effect, but only a small percentage of ABs against BoNT/A will be neutralizing, when specific epitopes of the BoNT/A molecule are attacked [27,28]. Therefore, the total abolishment of the clinical effect of BoNT/A after the first injection in the sense of complete secondary treatment failure is rare, but may occur [5] in different indications.

In clinical practice it will usually take time, up to years, until a treating physician and/or the patient will realize that BoNT/A therapy does not work as well as in the beginning of the BoNT/A therapy. This delay between the onset of NAB formation and the realization of a possible secondary treatment failure depends on the dose per session, the durations of the treatment cycles and the sensitivity of the patient and physician to a reduction in the efficacy of BoNT/A therapy. Therefore, the detection of a NAB-induced partial secondary treatment failure is a challenging problem in clinical practice. In this case, it is useful to have documents on the temporal development of the course of the severity of disease at hand to compare how effective previous BoNT/A injections have been. Moreover, the easy-to-handle 3- or 5-point scale (+2 much better, +1 better, 0 no change, -1 mild worsening, -2 moderate to severe worsening) is not sensitive enough to detect the slow development of PSTF in a patient. 

Even in cross-sectional studies, it may occur that a patient with an excellent clinical response (TSUI = 0, Figure 1B) has a NAB titer which can be detected in the MHDA [8] and the clinical outcome may be unsatisfactory, but the MHDA test is negative or close to threshold (Figure 1B,C) [29]. The correlation between the paralysis time (the outcome measure of the MHDA-NAB test) and the clinical outcome measured by means of the TSUI score in 39 patients with a positive AB-ELISA test was significant, but showed a large variability and scattering of the result [12]. About 50% of the patients with a clinical PSTF had a negative MHDA test [29,30]. However, this does only imply that the MHDA is not sensitive enough to detect NABs in those patients. It does not imply that no NABs are present in those patients.

A clear hint for the presence of a NAB-induced PSTF is a systematic reduction in the duration of efficacy [3], corresponding to a systematic worsening of symptoms at the end of an injection cycle as long as the duration of the injection cycle is kept constant [4]. In case a patient asks for shorter cycle durations, although the dose has already been increased, a PSTF has to be conjected. A full blown complete STF cannot be overlooked. Fortunately, CSTF is rare and usually develops over many treatment cycles [5].

### 3.2. Clinical Hints for the Presence of NABs in a Cohort

In clinical practice, the dose per session is well documented in a botulinum toxin outpatient clinic. In several studies, an increase in dose with the duration of treatment has been reported. When 212 CD patients were split up according to the results of an ELISA screening test for the presence of Abs, a steeper increase in dose with the duration of treatment was found in the 39 ELISA-positive compared to the ELISA-negative patients. The paralysis time of the MHDA of the 39 ELISA-positive patients significantly correlated with the mean dose per session. This indicates that the increase in dose during treatment is not only an indicator for an attempt to optimize the outcome, but also an indicator for the development of NABs [8,9,10,12].

The occasional occurrence of PSTF in a cohort during ongoing BoNT/A treatment does not necessarily lead to a significant change in the mean outcome measures [8,22]. However, the influence of these patients on outcome appears to be strong enough to keep the standard deviation of the outcome constant, or even to increase the variability of the outcome (Figure 3). 

For the optimization of the outcome in a patient, it would be really helpful for the treating physician to have the all the individual’s course of the severity of the disease entity and the “normal range” of response for this disease entity available. This allows to decide whether the patient responds normally and whether they stay within the normal limits or crosses percentiles. This method was used to control the normal development of a baby. Similarly, it can be controlled whether BoNT/A therapy develops normally.

### 3.3. The Influence of Protein Load of a BoNT/A Preparation on NAB Induction

The lesson learnt from the transition from “the old” to the “new Botox” is that the protein load of a BoNT/A preparation has a major influence on the antigenicity of a preparation. After the clinical experience that repetitive applications of the “old” Botox^®^ led to the frequent occurrence of immunoresistance against this preparation, the preparation was purified and the protein content substantially reduced. This procedure reduced the risk of NAB induction by a factor of 6 when the “old” and the “new” onabotulinumtoxin type A (onaBoNT/A) were compared [14,15,16]. The recommendation in 2001 was “to use that BoNT/A preparation with the lowest protein content for optimization of BoNT/A therapy” [17].

We think that this recommendation still holds, especially after the license of incoBoNT/A in 2005. The clostridial protein content of incoBoNT/A is 10-fold lower than that of abo- or onaBoNT/A [19]. Moreover, the neurotoxin protein content per 100 units is lower in the incoBoNT/A preparation than in the other two BoNT/A preparations [18]. This means that these two preparations contain fragments of the neurotoxin type A molecule which are not biologically but possibly immunologically active [19]. 

It has been reported that even in long-term treated patients, no patient was tested positive who had exclusively been treated with incoBoNT/A [2]. Animal experiments also indicate that the antigenicity of incoBoNT/A might be lower than that of the other two BoNT/A preparations [31]. 

### 3.4. Best Outcome Is Different between the Three BoNT/A Preparations

When BoNT/A is reinjected before the efficacy of the previous injection has fully declined, a staircase-like improvement occurs (Figure 1). When reinjections are performed at fixed intervals every 12 to 13 weeks, the mean outcome (measured at the end of the injection cycle) of the entire cohort will improve systematically. The question arises after how many injections or days of treatment an individual patient reaches their best result. In Figure 5, we compared the individual best results and times to the best results in a cohort of patients who were exclusively treated with incoBoNT/A (Figure 5B) and the patients who developed a PSTF later on and started their BoNT/A treatment with the complex protein containing a BoNT/A preparation abo- or onaBoNT/A (Figure 5A). Compared to the patients on incoBoNT/A-monotherapy, the patients on abo- and onaBoNT/A needed significantly more time and injections and did not reach an as good “best result” as the incoBoNT/A-treated patients. This is in full agreement with the hypothesis that NAB induction occurs early in the course of treatment and that the clinical effect is reduced early in the course of treatment. It is a strong argument not to start BoNT/A therapy with a complex protein containing BoNT/A preparation, but to use the most purified BoNT/A preparation with the lowest protein content from the very beginning [2,17,19].

## 4. Conclusions

This cumulative study mainly highlighted the importance of the systematic documentation of BoNT/A dose, injection intervals and cycles, as well as the valid objective and semi-quantitative disease severity scores, such as CD TSUI score to detect PSTF at the earliest time and to reduce the gap time between the onset of NAB formation and the realization of possible secondary treatment failure. 

Testing for NABs in line with clinical scoring is also helpful, not only in studies with a short period of observation, but also in longitudinal and long-term investigations.

The differences in the protein content of BoNT/A preparations have a crucial role in the incidence of NAB development and also treatment outcomes. NAB induction is more likely, when a preparation with a higher protein content is used. The best outcome in patients who develop a PSTF later on is worse than in patients who were exclusively treated with a low-protein-content preparation.

## 5. Materials and Methods 

This perspective was based on data from our research pool, which were gathered and documented for more than 20 years in our BoNT outpatient clinic of the university hospital of Düsseldorf (UKD) (HHU). This involved mainly collecting, rearranging and comparing the data from two cross-sectional studies, and finally summarizing the main clinical points that could be useful not only for daily practice, but also for the establishment of routine injection visits with all the necessary measurements and documentations to optimize the BoNT/A treatment efficacy.

All patients gave written informed consent. These two studies were performed according to the guidelines of good clinical practice (GCP) and approved by the local ethics committee of the University of Duesseldorf (study number: 4085-permission date for using the patient’s data in all related studies in the BoNT clinic of UKD in terms of prospective and retrospective studies: 05.Apr.2013) in accordance with the declaration of Helsinki.

### 5.1. The Cross-Sectional Study of 212 CD Patients

This study consisted of all patients with idiopathic CD who had been treated at the botulinum toxin out-patient clinic with BoNT-injections at least 10 times every 3–4 months (without disruption for the past 2–3 years) and who still experienced a treatment effect. 

The patients underwent a clinical examination just before the blood samples were taken for NAB determination. Besides the demographic data (age (61+/-11.8), gender (f/m 128/84), body weight (75.2+/-17.9), age at onset of CD 43.1+/-11.1)), treatment-related data (duration of treatment, scoring of the severity of CD by means of the TSUI score) [23] were determined by the attending physician. The patient’s subjective impression of the remaining severity of CD at time of the examination compared with the severity of CD just before the onset of BoNT-therapy was rated on a visual analogue scale (VAS 0–100; 0  =  no more symptoms, 100  =  CD severity just as bad as before the start of BoNT-therapy).

The patients were treated with abo-, ona-, inco- and rimabotulinumtoxinB (rimaBoNT/B). To allow comparison, the doses were transformed to ‘unified dose units’ (uDU): since most of the patients had been treated with aboBoNT/A, the doses of onaBoNT/A and incoBoNT/A were multiplied by three, the doses of rimaBoNT/B were divided by 10, and the aboBoNT/A-doses remained unchanged [32].

After all the patients were examined, all the serum samples were sent off together. The samples were first sent to determine the presence of antibodies using enzyme-linked immunosorbent assay (ELISA)-testing (fluoroimmunoassay). Thereafter, neutralizing the antibody titers of ELISA-positive samples was determined by means of MHDA. [8]

### 5.2. The Cross-Sectional Study of STF-CD Patients in Comparison to Patients under Monotherapy with IncoBoNT/A 

In the second cross-sectional, retrospective study, 63 CD patients with secondary treatment failure who had the following criterion, were included: 1. satisfactory pre-treatment with abo- or onaBoNT/A according to the patient´s assessment and with a response better than 2 points on the TSUI score by the physician´s assessment; 2. systematic worsening of the TSUI score over 2 treatment cycles of more than 2 TSUI score points before the switch to incoBoNT/A and the report by the patient of lack of efficacy during these 2 cycles; and 3. continuous incoBoNT/A treatment without interruption of more than one treatment cycle. In parallel, 32 CD patients who had exclusively been treated with incoBoNT/A without an interruption of more than one treatment cycle, were recruited (inco group). 

### 5.3. Statistical Analysis

In the first study, the data analysis was based on the 212 CD patients with known antibody status with stratification into the following subgroups: group I contained all ELISA-negative patients (n  =  173), and group II contained all ELISA-positive patients (n  =  39). This group contained eight MHDA-negative patients and 31 MHDA-positive patients; however, the paralysis times of the eight MHDA-negative patients were close to the threshold level of 2.31 mU/mL set by the Toxogen^®^ laboratory [12].

All the statistical analyses were carried out with the commercially available SPSS package (version 25: IBM, Armonk, NY, USA). After an ANOVA had yielded the differences among the subgroups in a first step, the comparisons between the subgroups were then performed nonparametrically using the Kendall tau B test. Results were confirmed by *t* test. Group size and parameters used always allowed the use of the *t*-test. Both nonparametric and parametric testing yielded the same significant results (with slightly different levels of significance). The Pearson correlation coefficient was used for correlation analysis [12].

In the second study, switchers (SWI group) were split up according to the pre-treatment: the ABO group comprised all those patients who had initially been treated with aboBoNT/A, and the ONA group with all those with initial onaBoNT/A treatment. Furthermore, the AK-pos group contained all the MHDA-positive and the AK-neg group all the MHDA-negative patients.

A one-way ANOVA with repeated measurements was calculated to compare the outcome measures ITSUI, BTSUI, STSUI, and ATSUI. For the comparison of the repeated measurements Greenhouse-Geisser tests were used. Furthermore, two to three group ANOVAs were performed to compare the ABO and ONA group and XEO-Mono group and to compare the AK-pos and AK-neg group and XEO-Mono group, respectively [33].

## Figures and Tables

**Figure 1 toxins-12-00499-f001:**
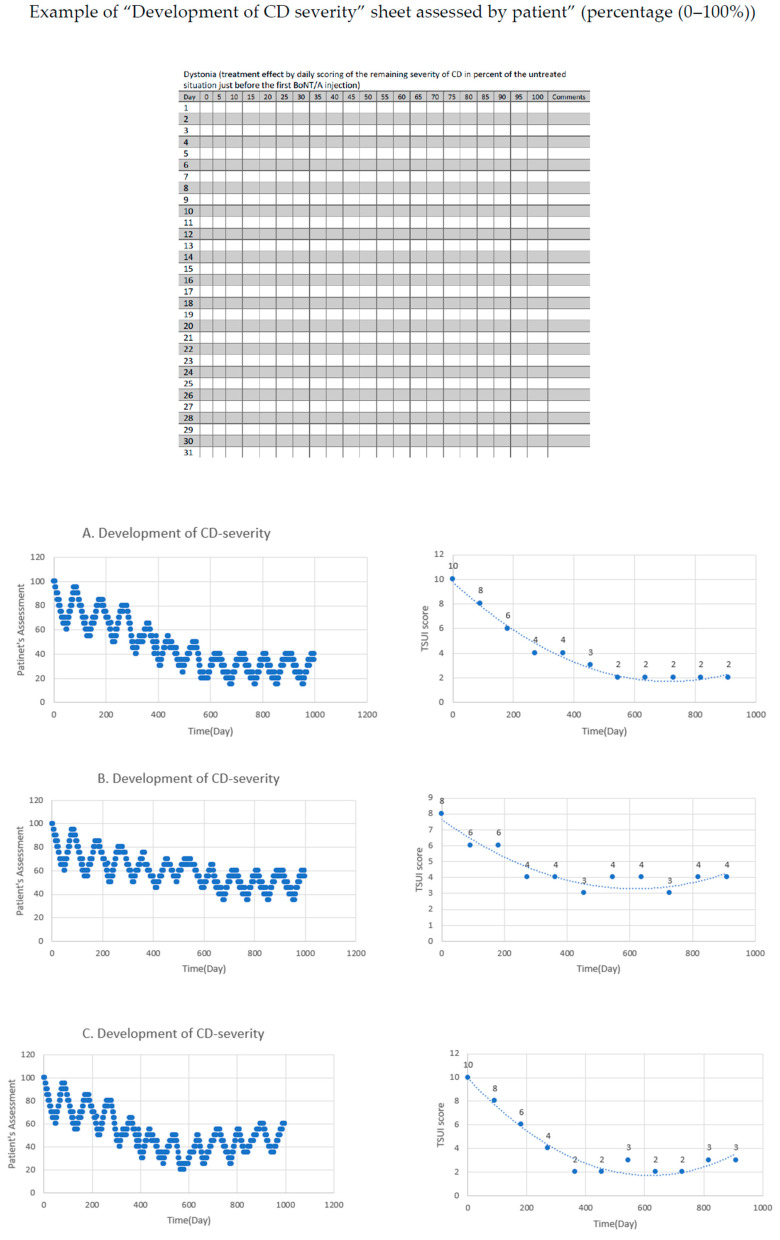
Development of cervical dystonia (CD) severity assessed by patient and physician (TSUI score): A–D, Efficacy of BoNT/A treatment and assessment of CD severity by patient (percentage (0–100%)) and physician (TSUI score) over time. (**A**): CD patient with the gradual improvement of symptoms to 25% and a TSUI score from 10 to 2 + mouse hemidiaphragm test (MHDA)-negative incobotulinumtoxin (incoBoNT/A). (**B**): CD patient with initial gradual symptoms’ improvement to 50% and a TSUI score from 8 to 4 + MHDA-positive with a moderate titer onabotulinumtoxin (onaBoNT/A) (**C**): CD patient with very mild secondary worsening + MHDA-positive with low titer abobotulinum (aboBoNT/A) (**D**): CD patient with secondary treatment failure + MHDA-positive with high titer (aboBoNT/A).

**Figure 2 toxins-12-00499-f002:**
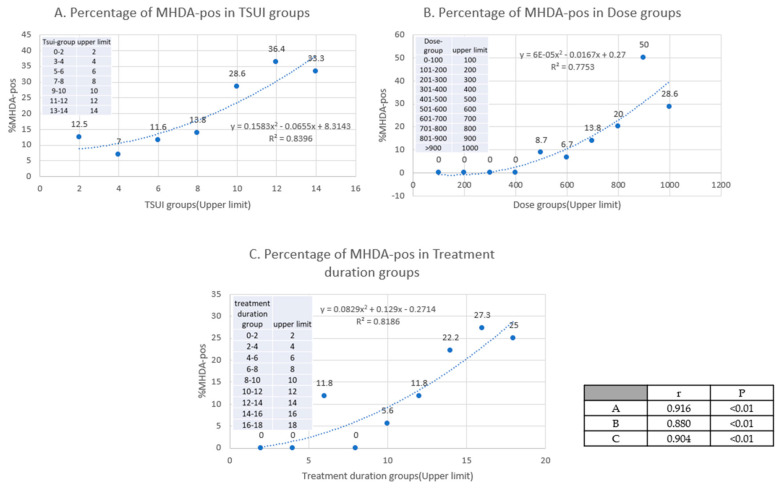
Percentage of MHDA-positive patients in the different groups (TSUI/Dose(uDU)/Treatment duration(years)). (**A**): in TSUI groups; (**B**): in Dose groups; (**C**): in Treatment duration groups.

**Figure 3 toxins-12-00499-f003:**
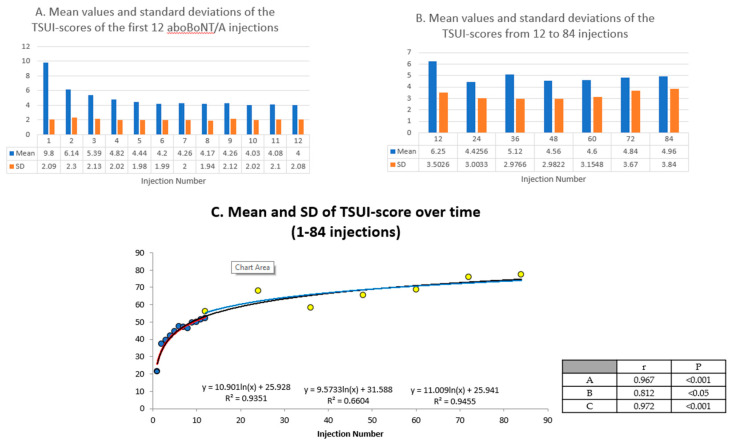
Mean values and standard deviations of the TSUI scores over time (during 1–84 injections). (**A**): In the first 12 injections (Kessler et al. study 1999); (**B**): from 12 to 84 injections (Hefter et al. study 2016); and (**C**): the combination of A and B over time (1–84 injections).

**Figure 4 toxins-12-00499-f004:**
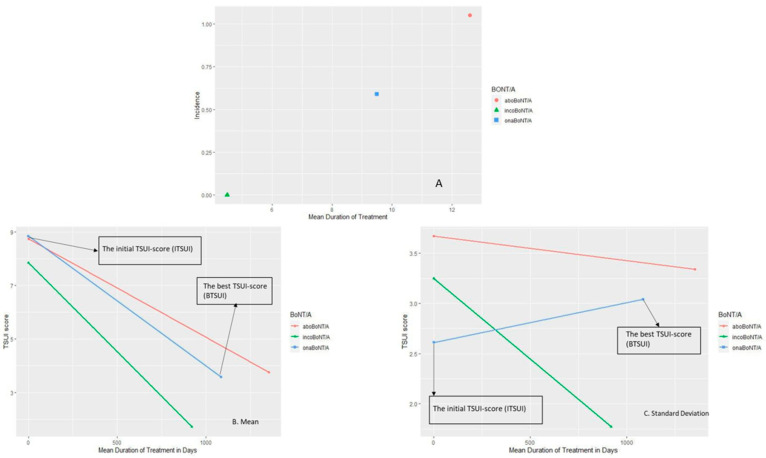
A–C, (**A**): Incidence of the MHDA-positive patients under abo, ona, incoBoNT/A monotherapy; (**B**): the development of the TSUI score mean from ITSUI to BTSUI over time; and (**C**): the development of the variability (SD) of the ITSUI and BTSUI over time.

**Figure 5 toxins-12-00499-f005:**
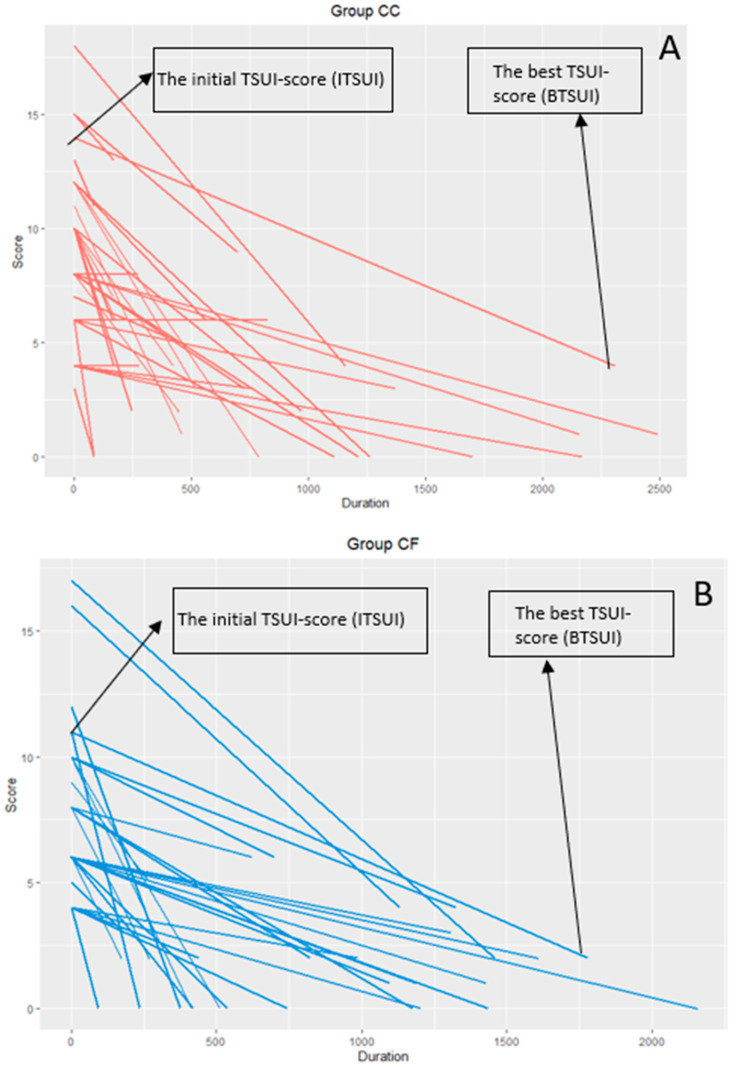
(**A**): Best outcome (BTSUI) in the CC group (ona- +abo group); and (**B**): best outcome (BTSUI) in the CF (inco group).

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
