# Peer review of "Clinical Implications of Difference in Antigenicity of Different Botulinum Neurotoxin Type A Preparations: Clinical Take-Home Messages from Our Research Pool and Literature"

_toxins, 2020, doi:10.3390/toxins12080499_

Round 1
Reviewer 1 Report
The manuscript entitled “Clinical implication of difference in antigenicity of different botulinum neurotoxin type A preparations: clinical take-home messages from our research pool” appears as well written and comprehensive. In my opinion it can be published in the present status.
Author Response
Thank you so much for your time and positive opinion on our manuscript.
Reviewer 2 Report
Clinical implications of difference in antigenicity of different botulinum neurotoxin type A preparations: clinical take-home messages from our research pool
The authors proved some arguments out of own research pool and from the literature to demonstrate that the clinical response and the antigenicity are different for the three BONT-A preparations.
Nevertheless, this manuscript need a very substantive correction and improvement before publishing is possible.
General points:
Please correct your title to: Clinical implications of difference in antigenicity of different botulinum neurotoxin type A preparations: clinical take-home messages from our research pool and literature.
Please add the list of abbreviations
Please add the Statistical analysis section to your manuscript.
Please check and correct all spaces between the words in the whole your manuscript.
Special points:
First of all, please add to your publication as a Supplement as a Table all information about the similar publications, using different types of BoNT-A in patients with CD.
Second, please describe and discuss all these results (about the similar publications, using different types of BoNT-A in patients with CD) in your publication and not only 3 publications. Otherwise, from my opinion, this publication provides only partial the information about the difference in antigenicity of different botulinum neurotoxin type A preparations.
Keywords
Please add also to keywords: botulinum toxin typ A; cervical dystonia;
Introduction
Lines 26-27: please add more the references at the end of this sentence.
Lines 27-34: please add the references at the end of all these sentences.
Lines 36-37: please add the references at the end of this sentence.
Line 39: please write out NABs
Lines 40-42: please add the references at the end of this sentence.
Lines 48-50: please add the references at the end of this sentence.
Line 49-50: please add the references at the end of this sentence.
Line 69: Please say: cervical dystonia (CD).
Results
Lines 73-75: you said: When BoNT/A therapy is started in a patient in our department, we instruct the patient to assess treatment effect by daily scoring of the remaining severity of CD in percent of the untreated situation just before the first BoNT/A injection (Fig. 1).
Please describe very exactly, how the patients assessed the treatment effect?
Figure 1: please add to γ axis: in which values you measured the Patient’s Assessment?
Lines 127-139: why used specifically only this publication from Kessler et al. [24] and not another also same and important publications from the same field?
Lines 185-194: please add references at the end of all these sentences.
Line 211: please add references at the end of this sentence.
Material and Methods
Lines 274-275: you said: This perspective is based on data from our research pool, which was gathered and documented more than 20 years in our BoNT outpatient clinic. Please add exactly your clinic name.
Lines 283-285: please add very exactly the patient’s information: gender, age, body weight, age of onset of CD.
What about the permission for your study? Please add the instance name, permission name and permission date for your experiments.
What about the permissions from each patient? Please add these whole information to your manuscript.
Line 290: you said: Patients had been treated with abo, ona, inco and rimabotulinumtoxinB (rimaBoNT/B).
Please add the exactly number of the patients, gender and age received each BoNT-A type: abo, ona, inco and rimabotulinumtoxinB (rimaBoNT/B).
Lines 300-308: please add very exactly the patient’s information: gender, age, body weight, age of onset of CD for both 63 CD-patients and 32 CD-patients studies.
Author Response
Thank you so much for your time and positive opinion on our manuscript. The following points were improved in our submitted manuscript:
1-title of our manuscript to “….and literature”.
2-the list of abbreviations, statistical analysis and suggested keywords were added at the end of the manuscript.
3-all spaces were checked and improved.
4- the position of all references were corrected based on the journal instruction (after each related sentences and before “,” , “.” …
5- Development of CD-severity assessed by patient was already explained in material and method but we also added the range of 0 to 100% to the text.
And for the Y axis of figure 1, the value of patient’s assessment was added in the title and also legend of figure (percentage (0-100%).
“Patient’s subjective impression of the remaining severity of CD at time of the examination compared with severity of CD just before onset of BoNT-therapy was rated on a visual analogue scale (VAS 0–100; 0 = no more symptoms, 100 = CD severity just as bad as before start of BoNT-therapy).” (291-294)
6-The name of our university hospital, responsible ethics committee and also patient’s informed consent statement were all added to the text.
7- the age, gender ratio, body weight, …. Were added for only first study but not for second one, the second manuscript is not out for publication at the moment and this manuscript is just about to announce the first interesting result of that.
8-The Kessler study was the best match for using TSUI score as an evaluating scale for CD and determination of TSUI mean value and standard deviation for each injections.
9-Lastly, we added a table of most recent articles about all 3 different BoNT/A preparations in comparison (from 2016 after Fabri et al. meta-analysis), in addition to that we are writing at the moment a new special meta-analysis with all articles including all 3 BoNT/A preparations.
Fabbri M, Leodori G, Fernandes RM, Bhidayasiri R, Marti MJ, Colosimo C, et al. Neutralizing Antibody and Botulinum Toxin Therapy: A Systematic Review and Meta-analysis. Neurotoxicity research. 2016;29(1):105-17.
10-All minor typing errors were corrected (cervical dystonia (CD)-NAB…)
Thank you so much

Reviewer 3 Report
This seems to be a properly designed and executed cross-sectional study, with reliable existing data from past clinical research and postmarketing surveillance records, applied the right analyzing tools and meaningful interpretation. The conclusions will surely help clinicians and patients make better choices for the relevant disorder or disease treatment, esp. when multiple approved brand exists in the market for dealing with the same illness. It should be published timely after minor corrections, such as line 57 on page 2 what is the correct number ? "4,87 ng per vial" should be 4.87 or 487 or another number?
Author Response
Thank you so much for your time and positive opinion on our manuscript. The mentioned typing error was corrected in text and the whole text was checked again for typing errors carefully.
Round 2
Reviewer 2 Report
Thank you, this manuscript was impressive corrected.
Unfortunately, the authors have not considered all of my previously suggestions.
Therefore, this manuscript need once more correction and improvement before publishing is possible.
As proposed already before, please add to your Introduction section: please describe and discuss all these results (about the similar publications, using different types of BoNT-A in patients with CD) in your publication and not only 3 publications. Otherwise, from my opinion, this publication provides only partial the information about the difference in antigenicity of different botulinum neurotoxin type A preparations.
Introduction
As already before proposed from me:
Lines 27-28: please add more the references at the end of this sentence.
Lines 28-35: please add the references at the end of all these sentences.
Lines 36-38: please add the references at the end of this sentence.
Lines 41-43: please add the references at the end of this sentence.
Lines 49-51: please add the references at the end of this sentence.
Results
As already before proposed:
Lines 74-76: you said: When BoNT/A therapy is started in a patient in our department, we instruct the patient to assess treatment effect by daily scoring of the remaining severity of CD in percent of the untreated situation just before the first BoNT/A injection (Fig. 1).
Please describe very exactly in your manuscript, how the patients assessed the treatment effect?
Discussion
As proposed already before:
Lines 186-195: please add references at the end of all these sentences.
As proposed already before:
Line 212: please add references at the end of this sentence.
Material and Methods
As proposed already before:
What about the permission for your study? Please add the permission name and permission date for your experiments.
Author Response
Thank you so much for your time and very careful review.
As proposed already before, please add to your Introduction section: please describe and discuss all these results (about the similar publications, using different types of BoNT-A in patients with CD) in your publication and not only 3 publications. Otherwise, from my opinion, this publication provides only partial the information about the difference in antigenicity of different botulinum neurotoxin type A preparations.
Reply---as we wrote in last revision response, all articles were included in this manuscript (till 2016- just Fabri et al. meta-analysis/ after 2016- we prepared a table of references as a supplementary file) and the studies of Walter U, et al./ Hefter H, et al. (3) / Dressler D, et al./ Albrecht P, et al./ Kamm C, et al./ Frevert J, et al. (3) were already cited. The study of Pan L, et al is also added in this version. And this is 15 different articles plus one meta-analysis till 2016 included all studies before 2016. If you have suggestions in terms of special title or articles related to this topic that we did not cover in this manuscript we would like to be informed.
Introduction
As already before proposed from me:
Lines 27-28: please add more the references at the end of this sentence.
Lines 28-35: please add the references at the end of all these sentences.
Lines 36-38: please add the references at the end of this sentence.
Lines 41-43: please add the references at the end of this sentence.
Lines 49-51: please add the references at the end of this sentence.
Discussion
As proposed already before:
Lines 186-195: please add references at the end of all these sentences.
As proposed already before:
Line 212: please add references at the end of this sentence.
Reply---all references were corrected and relocated to the end of sentences.
For lines 27-28: There is one guideline, that means we need no more articles or references related to this part of our manuscript.
Simpson DM, Hallett M, Ashman EJ, Comella CL, Green MW, Gronseth GS, et al. Practice guideline update summary: Botulinum neurotoxin for the treatment of blepharospasm, cervical dystonia, adult spasticity, and headache: Report of the Guideline Development Subcommittee of the American Academy of Neurology. Neurology. 2016;86(19):1818-26.
Results
As already before proposed:
Lines 74-76: you said: When BoNT/A therapy is started in a patient in our department, we instruct the patient to assess treatment effect by daily scoring of the remaining severity of CD in percent of the untreated situation just before the first BoNT/A injection (Fig. 1).
Please describe very exactly in your manuscript, how the patients assessed the treatment effect?
Reply---the description was completed and also one figure was added for more clarification.
When BoNT/A therapy is started in a patient in our department, we instruct the patient to assess treatment effect by daily scoring of the remaining severity of CD in percent of the untreated situation just before the first BoNT/A injection on a visual analogue scale (VAS 0–100; 0 = no more symptoms, 100 = CD severity just as bad as before start of BoNT-therapy)(Fig.1).(result)
Patient’s subjective impression of the remaining severity of CD at time of the examination compared with severity of CD just before onset of BoNT-therapy was rated on a visual analogue scale (VAS 0–100; 0 = no more symptoms, 100 = CD severity just as bad as before start of BoNT-therapy). (291-294)(material and methods)
Material and Methods
As proposed already before:
What about the permission for your study? Please add the permission name and permission date for your experiments.
Reply--- permission date and name were added.
All patients gave written informed consent. These two studies had been performed according to the guidelines of good clinical practise (GCP) and approved by the local ethics committee of the University of Duesseldorf (study number: 4085-permission date for using patient’s data in all related studies in BoNT clinic of UKD in terms of prospective and retrospective studies: 05.Apr.2013) in accordance with the declaration of Helsinki.
Round 3
Reviewer 2 Report
Thank you, this manuscript was corrected according to my proposals.
Author Response
Thank you so much for your very careful comments, which made our work so much better and more useful to readers.